# Development of Rating Curves: Machine Learning vs. Statistical Methods

**Evangelos Rozos [1],\*, Jorge Leandro [2] and Demetris Koutsoyiannis [3]**

[1] Institute for Environmental Research & Sustainable Development, National Observatory of Athens, 15236 Athens, Greece

[2] Research Institute for Water and Environment, Faculty IV School of Science and Technology, University of Siegen, Paul-Bonatz-Str. 9–11, 57068 Siegen, Germany

[3] Department of Water Resources and Environmental Engineering, School of Civil Engineering, National Technical University of Athens, 15780 Athens, Greece

\* Correspondence: erozos@noa.gr; Tel.: +30-210-810-9125

**Abstract:** Streamflow measurements provide valuable hydrological information but, at the same time, are difficult to obtain. For this reason, discharge records of regular intervals are usually obtained indirectly by a stage–discharge rating curve, which establishes a relation between measured water levels to volumetric rate of flow. Rating curves are difficult to develop because they require simultaneous measurements of discharge and stage over a wide range of stages. Furthermore, the shear forces generated during flood events often change the streambed shape and roughness. As a result, over long periods, the stage–discharge measurements are likely to form clusters to which different stage–discharge rating curves apply. For the identification of these clusters, various robust statistical approaches have been suggested by researchers, which, however, have not become popular among practitioners because of their complexity. Alternatively, various researchers have employed machine learning approaches. These approaches, though motivated by the time-dependent nature of the rating curves, handle the data as of stationary origin. In this study, we examine the advantages of a very simple technique: use time as one of the machine learning model inputs. This approach was tested in three real-world case studies against a statistical method and the results indicated its potential value in the development of a simple tool for rating curves suitable for practitioners.

**Keywords:** stage–discharge relationship; rating curve; machine learning; multilayer perceptron; unsupervised learning; clustering; DBSCAN

## 1. Introduction

Precipitation and discharge measurements are fundamental data for hydrological applications [1]. Precipitation measurements provide representative information only for the area surrounding the rain gauge, with the extent of this area defined by the geomorphological characteristics. On the other hand, streamflow measurements provide information regarding the whole upstream area of the cross-section where the measurements take place. This makes evident the importance of streamflow measurements to hydrological studies and research.

Precipitation measurements can be easily obtained by installing, for example, on a mast, a rain gauge. However, streamflow measurements are not trivial to obtain since they require velocity sampling at multiple locations of a cross-section and the integration of these measurements over the entire cross-sectional area. For this reason, streamflow is measured, most of the time, indirectly via the stage, which is transformed into discharge with a stage–discharge relationship [2], usually a power equation [3], also known as rating curve. Unfortunately, the stage–discharge relationship can vary with time, in response to

degradation, aggradation, or a change in the channel shape at the control section [3]. This results in a shift in the measurements arriving after each cross-section change (different discharge for the same stage). As a consequence, the stage–discharge measurements tend to cluster into groups that correspond to different hydraulic behaviours of the cross-section.

The partitioning of data into clusters, in the sense of hydrologic similarity, is a pre-processing task involved in many applications [4], such as clustering analysis of rainfall time series [5,6], grouping of streamflow regimes [7], hydro-geomorphometric characterisation of watersheds [8], etc. Specifically, for the stage–discharge measurements, various statistical techniques have been employed by researchers for dividing the time series into segments. For example, Zhou et al. [9] applied a statistical method that employs the t-test to identify the changes in the mean value. However, the complexity of this method is $O(N^N)$ [10,11], which practically imposes a compromise between the time spent in calculations and the achieved acceptable precision. On the other hand, Tsakalias and Koutsoyiannis [12] proposed a sophisticated method, called PINAX, that considers multiple statistics in the analysis of the time series and can be completed within a few seconds. PINAX will be used as a benchmark method in this study.

The processing of stage–discharge measurements can be considered a data mining task, i.e., the extraction and discovery of patterns in datasets. The recent trend is to employ general-purpose machine learning tools for data mining, which also have the advantage of being simpler in their application. Indeed, there are plenty of studies that have employed machine learning in stage–discharge relationships. For example, Bhattacharya and Solomatine [13], motivated by the fact that "Stage and discharge are time-dependent and very often they exhibit random fluctuations, their relationship is not always unique.", employed a multi-layer perceptron (MLP) with four inputs (discharge of previous time step, stage of the current and two previous time steps). Al-Abadi [14] (with a similar motivation "… the functional relationship between stage and discharge is complex, time-varying…") evaluated three alternative machine learning methods with various setups to obtain the discharge from stage measurements. These three methods include: an MLP (similar to that used in [13]), an M5 decision tree, and a Takagi–Sugeno fuzzy model. Other researchers employed support-vector machines for the same purpose [15]. Londhe and Panse-Aglave [16] compared an MLP and an M5 decision tree against non-linear regression (actually an extension of the typical stage–discharge power law). They found that both MLP and M5 outperformed the non-linear regression. Jiang et al. [17] also tested a machine learning approach (*k*-means combined with *k*-nearest) against a statistical method (locally weighted regression) and also found that the former achieved superior performance. Finally, many researchers combined various machine learning approaches with wavelet transform (e.g., [18,19]).

The previous approaches appear to offer a more accurate estimation of the discharge. This improvement can be partly explained by the ability of the employed machine learning approaches to reproduce the hysteresis effect [18,19], which is manifested by the characteristic loop in the stage–discharge plots [15,20]. However, even though the previous studies mention as their motivation the time-dependent nature of the stage–discharge relationship, they did not take into account the non-periodic dependency of this relationship on time because of the changes in the cross-section geometry. In all studies, the machine learning model inputs are a combination of stages and discharges of consecutive time steps. These inputs, along with the wavelet transforms, may be capable of reproducing hysteresis loops, but it is doubtful that they can represent arbitrary changes in the cross-section.

In this study, we employed a simple technique to take into account the dependency of the stage–discharge relationship on time: let time along with stage be the machine learning model inputs. For the pre-processing of data (i.e., remove outliers), we employed unsupervised learning [21]. We used PINAX, a pure statistical tool, as a benchmark model.

Both PINAX and the suggested technique were applied to real-world case studies to draw conclusions on the advantages and disadvantages of the two methods.

## 2. Materials and Methods

### 2.1. The Statistical Approach—PINAX

PINAX attempts to partition the stage–discharge measurements by maximising a single objective function. This function combines statistical/conceptual consistency evidence (binary value), a measure of consistency, and the number of outliers:

$$f(\Delta, U) = \text{He}(\Delta)[\text{card}(U) + \text{Hm}(\Delta)], \tag{1}$$

where $\Delta$ is the partition of the dataset (the stage–discharge measurements), $U$ is the dataset without the outliers, $\text{He}(\cdot)$ is a binary function that expresses the statistical/conceptual consistency evidence, $\text{card}(\cdot)$ is a function that returns the cardinality of a set, and $\text{Hm}(\cdot)$ is the measure of consistency.

The measure of consistency is the conditional probability of the event $\{\underline{R}^2 < r^2 \,|\, \underline{R}^2 < \rho_0^2\}$, where $\underline{R}^2$ is the random variable of which the estimate is $r^2$, the determination coefficient (DC), and $\rho_0^2$ is a desirable lower threshold of the DC (e.g., $\rho_0^2 = 0.9$).

The consistency evidence function is the product of multiple binary functions (serving as the logical AND applied on them) which include:

- A test of the null hypothesis that the DC is equal to a selected value $\rho_0$ against the alternative hypothesis to be less than $\rho_0$ at a significance level $\alpha_1$.
- A test that the standardised departures from the regression estimations are less than an upper threshold (the acceptable threshold corresponds to a significance level $\alpha_2$).
- A test of the null hypothesis that the standardised deviation of the residuals is equal to a value $\sigma_0$ against the alternative to be greater with a significance level $\alpha_3$.
- An upper threshold for consecutive positive or negative residuals (the acceptable threshold corresponds to a significance level $\alpha_4$).
- An upper threshold for standardised departures of inputs (stage) and outputs (discharge) from their corresponding mean values (the acceptable threshold corresponds to a significance level $\alpha_5$).
- An upper threshold $b_6$ for the number of clusters.
- The known outliers and breakpoints.

It should be noted that Equation (1) puts emphasis on the number of outliers. That is, among two consistent solutions ($\text{He}(\Delta) \neq 0$), the solution with the fewer outliers results in a greater value of Equation (1) (the greater the better).

A blackboard system [22] is employed to maximise Equation (1). Blackboard systems are based on knowledge sources to transform the system state (the partitioning of stage–discharge measurements). The knowledge sources are organised in a list of strategies of transformation and the preconditions for their application. The strategies employed by PINAX are displayed in Figure 5 of [12].

### 2.2. The Machine Learning Approach

#### 2.2.1. Data Clustering and Filtering

Initially, we investigated whether a machine learning clustering method can be used instead of the knowledge sources of PINAX for the grouping and filtering of the available measurements. For this reason, 10 clustering methods were assessed on a dataset of real-world measurements (stage–discharge measurements of Sakoulevas River in Western Macedonia, Greece). The code used for this comparison was based on the code found in [23].

The dataset included 78 stage–discharge measurements. Each measurement corresponds to a 3D point of which the coordinates are the time, the stage, and the discharge. The time was encoded employing the MS Excel date format [24]. The stage and discharge were logarithmised. All coordinates were normalised with the z-score

normalisation [25]. The results of the 10 clustering methods are displayed in Figures A1 and A2. According to these figures, DBSCAN achieved the best performance.

Density-based spatial clustering of applications with noise (DBSCAN) is a non-parametric algorithm [26]. DBSCAN can discover clusters with arbitrary shape and is robust to outliers [27]. The algorithm of DBSCAN is quite simple. The points of an *n*-dimensional space are grouped into groups according to their spatial distribution. Assume the *k* points $p_i$, *i* = 1…*k*. The algorithm has two parameters: the distance $\varepsilon$, which defines the neighbourhood of each point, and a threshold *m* corresponding to the minimum number of points within the neighbourhood of a point $p_i$. The points with at least *m* points within their neighbourhood are called core points. A point $p_{i+1}$ which is within the neighbourhood of the core point $p_i$ but has less than *m* points within its neighbourhood is called directly reachable from core points. A point that has no neighbours is an outlier (Figure 1).

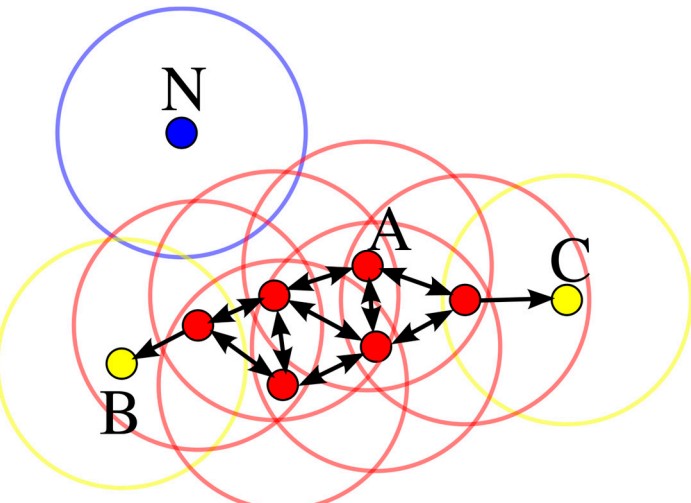

**Figure 1.** DBSCAN clustering algorithm, shared from [27] (CC BY-SA 3.0). N is an outlier, A is a core point, B and C are reachable from A but not core points. The radius of all circles (hyperspheres in an *n*-dimensional space) is $\varepsilon$.

According to the previous, DBSCAN has two parameters, *m* and $\varepsilon$. The threshold *m* was kept equal to 6 according to the ISO Standard 1100 for the determination of stage–discharge relationships [28]. Coincidently, this is equal to the recommended *m* value for applying DBSCAN to a 3-dimensional space (twice the dimensions) [29]. The parameter $\varepsilon$ was manually calibrated. We started from 1, then tried the values 0.5 and 1.5. After finding the direction that improved the performance, we repeatedly tried the $\varepsilon$ value in the middle of the best pair of $\varepsilon$ values of the previous try. Best $\varepsilon$ was found to be 0.69 for the Sakoulavas River case study.

### 2.2.2. Approximate the Rating Curve with MLP

In this study we assume the existence of a stage–discharge relationship valid for all time instances, which is described by Equation (2).

$$Q_t = F(S_t, t), \tag{2}$$

where $Q_t$ is the discharge at time step *t*, $S_t$ is the stage measurement at time step *t*, and *F* is the stage–discharge relationship.

To approximate *F*, various machine learning approaches can be used for the regression. SVR is a good option for low-dimensional problems but needs specific approaches to deal with high-dimensional problems [30]. On the other hand, MLPs have been proven a good choice in hydrological applications of synthetic time series analysis [31] and approximation of model errors [32]. For this reason, the MLP approach was

selected to approximate the time-variant stage–discharge relationship, i.e., the *F* of Equation (2).

The concept of this approach is to train an MLP with ($Q_t$, $S_t$) pairs, i.e., stage and discharge measurements, and then to apply it, as a representative of Equation (2), with the regularly recorded stage measurements to obtain the corresponding discharge. Though this MLP could be applied at times beyond the period of available discharge measurements, its reliability will be reduced since it will not have access to any information regarding the succeeding changes in the cross-section shape.

Various MLP topologies were tested in this study. In the first case study, the topology was kept minimal (only one hidden layer with 6 nodes) and the MLP was applied on data where the outliers have been previously filtered by DBSCAN. In the following two case studies, where more data were available, more complex topologies were used (two hidden layers with 40 nodes) and the MLP was applied directly to the original dataset (no filtering).

The MLP network was implemented with Cortexsys (a deep learning toolbox for MATLAB and GNU Octave), the cost function was the mean squared error, and it was trained with the gradient descent algorithm ADADELTA [33].

## 3. Results

### 3.1. Case Study—Sakoulevas River

The dataset consists of 78 stage–discharge measurements from Sakoulevas River, Greece. The measurements extend over an eight-year period, with the first one performed on 29 February 1964 and the last one on 26 January 1972.

The PINAX partitioned the available data into three periods and rejected 24 points as outliers. The application of a power–law trend line to the data of these three periods is displayed in Figure 2.

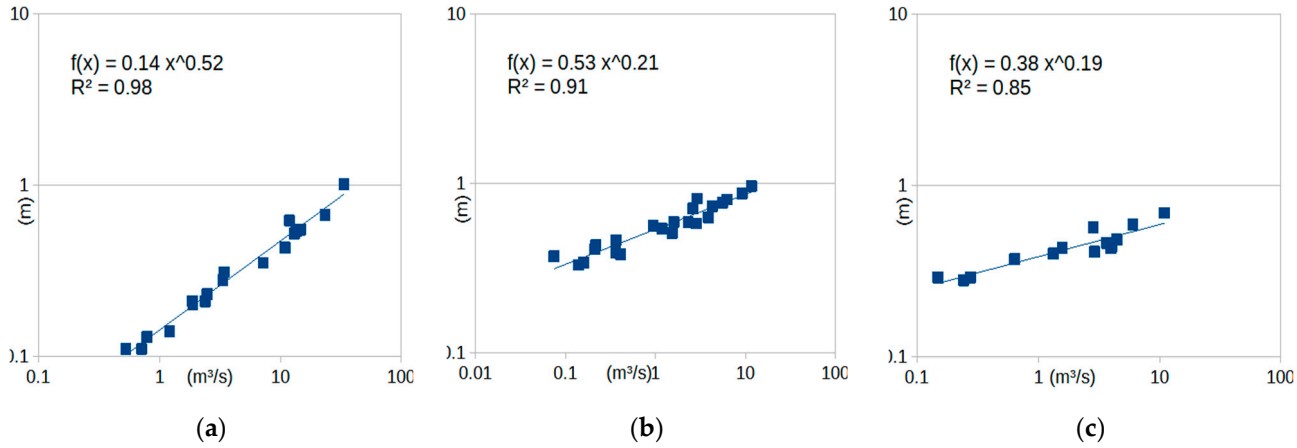

**Figure 2.** Sakoulevas case study—the fit of a power–law trend line to the data partitioned with PINAX: (**a**) period 1 (02/1964–05/1966); (**b**) period 2 (06/1966–05/1969); (**c**) period 3 (06/1969–01/1972).

The first stage to handle the same dataset with machine learning is to apply DBSCAN to filter the outliers. The following figures display the results of the pre-processing with DBSCAN, which detected two clusters of measurements and 14 outliers. For the majority of the points deemed outliers, their elimination can be explained by examining Figures 3–5.

- Points 3 and 5. It is evident from Figure 5 that these two points lie far away from the cloud of the remaining points.

- Point 4. This point corresponds to a measurement with very low discharge (the second lowest, see Figure 3), which, however, has a significant stage (Figure 4). This is the only point of DBSCAN that was not deemed an outlier by PINAX. Indeed, geometrically, it looks like an outlier, but from a hydraulic perspective, this judgement can be disputed.
- Points 9, 10, and 11. It is evident from Figures 3 and 4 that these points violate the expected monotonic relationship between stage and discharge (these points have higher discharge but lower stage from their neighbours).
- Point 14. It is evident from Figures 3 and 4 that this point violates the expected monotonic relationship between stage and discharge (in an opposite direction from the previous three points).
- Points 8, 12, and 13. These points could have formed a category on their own, but the lower limit to form a new category is 6 points (this was verified by setting *m* equal to 3).
- Points 1, 2, 6, and 7. The reason for this decision of DBSCAN is not obvious for these points.

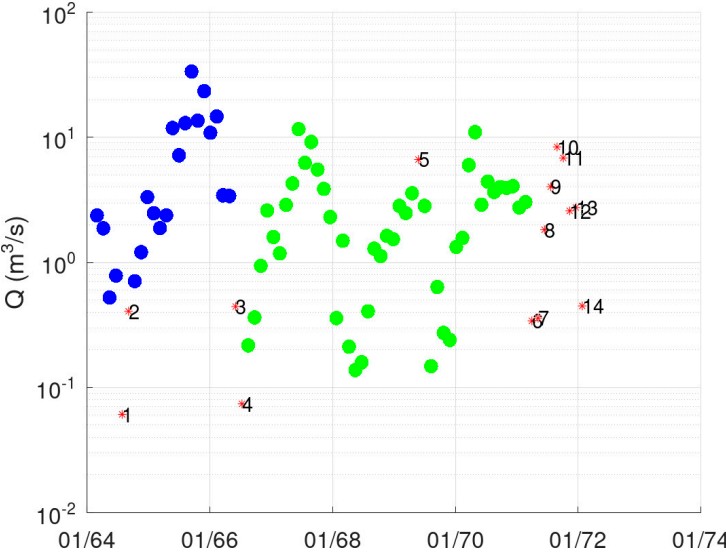

**Figure 3.** Sakoulevas case study—discharge time series plot. With different colours, the partition of DBSCAN. Period 1 (02/1964–04/1966) with blue dots, period 2 (05/1966–01/1972) with green dots, outliers with red stars.

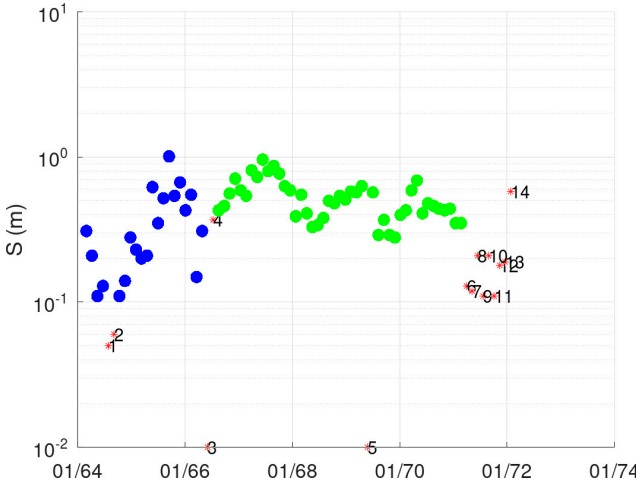

**Figure 4.** Sakoulevas case study—stage time series plot. With different colours, the partition of DBSCAN. Period 1 (02/1964–04/1966) with blue dots, period 2 (05/1966–01/1972) with green dots, outliers with red stars.

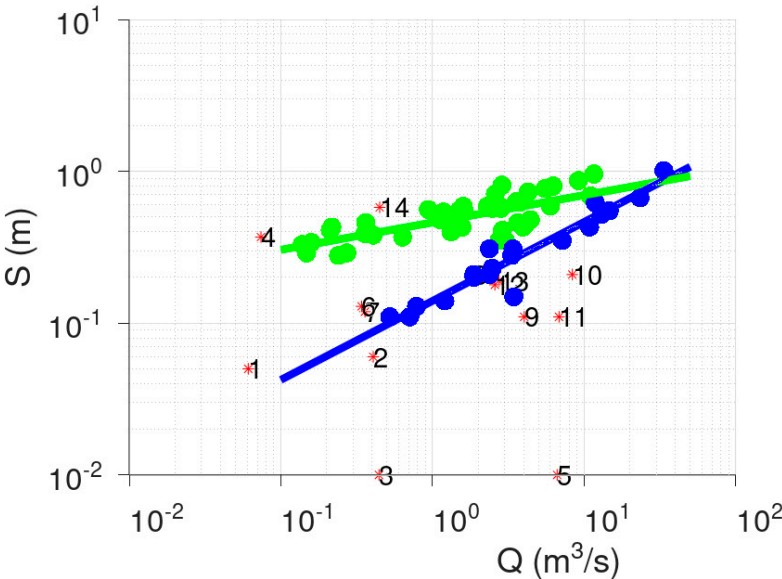

**Figure 5.** Sakoulevas case study—scatter plot of stage vs. discharge. With different colours, the partition of DBSCAN. Period 1 (02/1964–04/1966) with blue dots, period 2 (05/1966–01/1972) with green dots, outliers with red stars.

Despite the relatively good performance of DBSCAN, it fell short of the PINAX performance. It partitioned the dataset into two groups (instead of three groups suggested by PINAX), but the DC of the second group of points was very low (see Figure A3). For this reason, DBSCAN was used only at the pre-processing stage for identifying the outliers.

At a second stage, the MLP was trained using the dataset without the points that were deemed as outliers by DBSCAN. Three layers were used with 2-6-1 nodes. The input layer was fed with the pre-processed time and stage, the same time series used in DBSCAN, but without the points marked as outliers. The output layer gives the discharge. The activation function of the second layer was the sigmoid [34], whereas no activation was used for the output.

In order to have a fair comparison with PINAX, the number of nodes in the second layer of the MLP network was set equal to the value, resulting in similar degrees of freedom to that of PINAX. For this specific case study, PINAX partitioned the data into three groups. A power regression was applied to each group, which means $3 \times 2 = 6$ parameters. Further, the definition of these three groups requires two milestones. Finally, PINAX removed 24 points as outliers (each point can be identified by its id or rank in the data). Therefore, in order to describe the stage–discharge relationship obtained by PINAX, $6 + 2 + 24 = 32$ parameters are required. The number of parameters for the MLP was the number of weights and biases in the MLP network, 25, plus the number of outliers, 14, i.e., 39 parameters in total.

The grouping of DBSCAN was not taken into account at this stage. Figure 6 displays the calculated discharge from the trained MLP network and the calculated discharge from the three regressions obtained with PINAX compared against the available discharge observations. The outliers were not included in this plot.

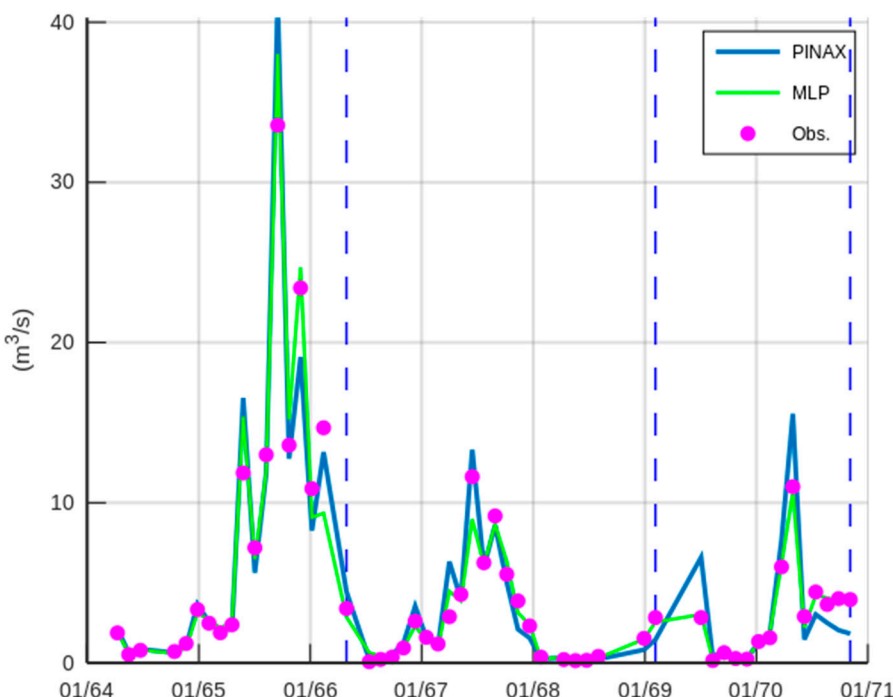

**Figure 6.** Sakoulevas case study—discharge calculated with the MLP, and discharge calculated with the 3 regression equations after partitioning data with PINAX. PINAX period 1 (02/1964–05/1966), PINAX period 2 (06/1966–05/1969), PINAX period 3 (06/1969–01/1972). The start and end of the periods are marked with the dashed vertical lines.

Figure 7 displays the comparison of the outliers obtained with DBSCAN and PINAX methods in the Sakoulevas case study. According to this figure, DBSCAN detected two clusters, whereas PINAX detected three. PINAX deemed 24 measurements to be outliers, whereas DBSCAN 14, of which 13 belong to the set of 24 of PINAX (Point 4 is not). Points 1, 2, 6, and 7 were deemed outliers by both algorithms, though the reason for this decision is not apparent.

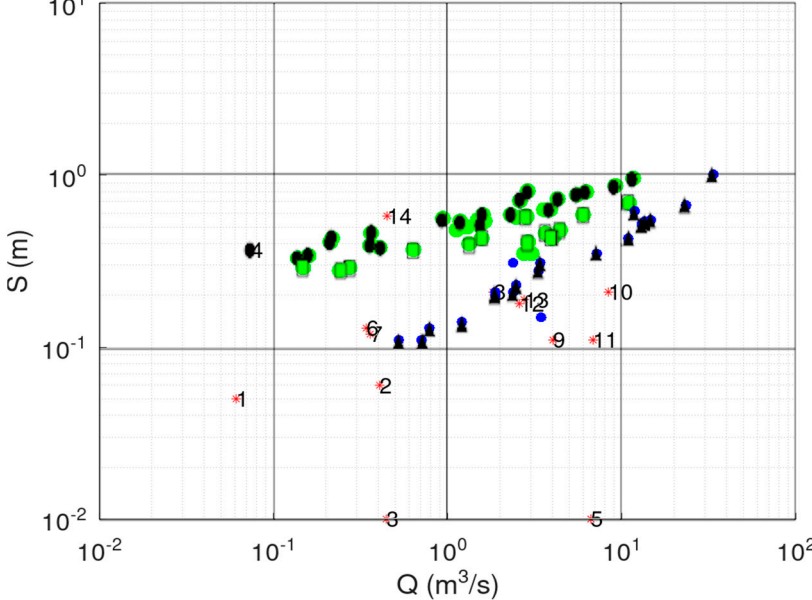

**Figure 7.** Sakoulevas case study—scatter plot of stage vs. discharge. DBSCAN: period 1 (02/1964–04/1966) with blue dots, period 2 (05/1966–01/1972) with green dots, outliers with red stars. PINAX:

period 1 (02/1964–05/1966) with triangles, period 2 (06/1966–05/1969) with black dots, and period 3 (06/1969–01/1972) with rectangles.

DBSCAN appears to be successful in detecting outliers. However, it is not very successful in partitioning this kind of hydrological data to allow for a traditional approach (i.e., first partition and then apply separately to each cluster a linear regression on double logarithmic data). For this reason, as mentioned previously, an MLP was trained with the filtered data (no outliers) to obtain an approximation of the stage–discharge relationship. The MLP was employed with two inputs, the stage and the time, in order to represent the time-dependent properties of the cross-section.

An intriguing use of the MLP comes when a steady stage is used as input (for example, the mean stage of the available measurements). In this case, the MLP returns the discharge of the cross-section for the same stage over different time instances of the assessed period. This is actually a schematic representation of how the cross-section change is perceived by the trained MLP. Figure 8 displays the discharge calculated by the MLP and the PINAX-obtained regressions for a constant stage equal to the mean stage. According to this figure, the two approaches present a similar schematisation of the cross-section change.

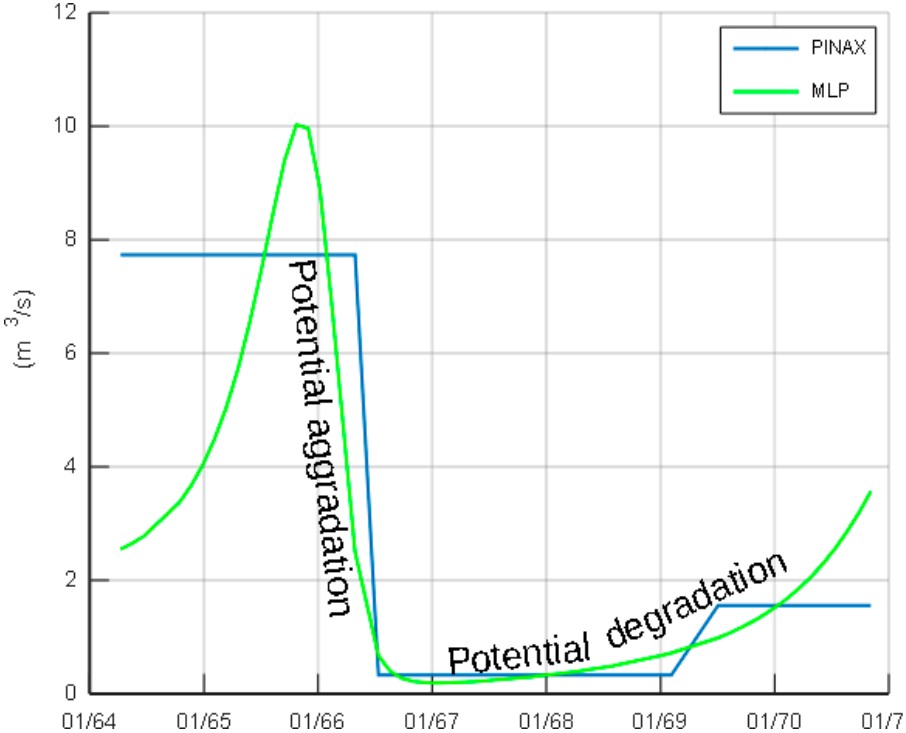

**Figure 8.** Sakoulevas case study—discharge over time corresponding to a constant stage (equal to mean observed stage) as it is calculated by the 3 PINAX-obtained regressions and the MLP.

Table 1 displays the statistical tests employed in the PINAX application [12]. Only the four first tests of the six mentioned in Section 2.1 were used. PINAX does not offer solutions that fail, even in a single test. The MLP fails at the second test despite having a better DC (0.97 instead of the overall 0.93 of PINAX). This is happening because, according to the second test design, the departures are divided by the standard deviation of residuals, which is much lower in the MLP than in PINAX (0.26 instead of 0.4 m$^3$/s). The scope of this test in PINAX was mainly to detect outliers and it does not have a significant impact on the overall performance of the method.

**Table 1.** The statistical tests of PINAX.

| Statistical Test | PINAX | MLP |
|---|:---:|:---:|
| (attained significance level of DC equal 0.9) > $\alpha_1 = 0.05$ | ✓ | ✓ |
| (standardised departures) < $b_2 = 2.58$ | ✓ | X |
| (standard deviation of residuals) < $\sigma_0 = 0.35$, with $\alpha_3 = 0.05$ | ✓ | ✓ |
| (number of consecutive runs) < 9 | ✓ | ✓ |

In order to test the contribution of the inclusion of time as an input to an MLP, an attempt to train a network with only the stage as input was made. The topology of this network was 1-20-1. The number of nodes in the hidden layer was increased to have at least the same number of parameters as the 2-6-1 network, hence, a similar model strength (actually, the 1-20-1 network has much more parameters than the 2-6-1 network, 61 instead of 39). However, the performance of the model was poor. The DC was only 0.48, which indicates that the inclusion of the time variable in Equation (2) helped significantly to improve the approximation accuracy of the stage–discharge equation. This finding was expected, since in a time-variant cross-section, a single stage value would correspond to multiple discharge values, different before and after each cross-section change. The time variable provides the means to distinguish between these values. This is the reason for the poor performance of the MLP without the time-variable input.

In the following two case studies, where more data were available, a more complex MLP topology was employed. The idea was to investigate whether a deep MLP network is capable of efficiently handling data without any pre-processing, i.e., without the DBSCAN grouping and filtering (Figures 3–5).

*3.2. Case Study—Trikeriotis River*

The dataset consists of 266 stage–discharge measurements from Trikeriotis River, Greece. The measurements extend over an eighteen-year period with the first one performed on 21 March 1972 and the last one on 2 October 1990. PINAX filtered 22 out of the 266 measurements. PINAX partitioned the available stage–discharge measurements into 14 periods. The DC of the combined corresponding 14 regressions was 0.961.

In this case study, an MLP was applied directly on the whole dataset. For this reason, a more complex network topology was used, aiming at capturing the patterns (including the detection of outliers) in the data. Two hidden layers were employed with 40 nodes in each layer. The activation functions were sigmoid–sigmoid–linear. The DC of the MLP for the whole dataset (the above DC value of PINAX is for the dataset without the outliers) was 0.96. The fit of the 14 PINAX-obtained regressions to the filtered dataset and of the MLP (actually serving as an approximation of Equation (2)) to the whole dataset is displayed in Figure 9.

Figure 10 displays the discharge calculated by the MLP and the PINAX-obtained regressions for a constant stage equal to the mean stage. According to this figure, the cross-section changes estimated by the two approaches are very similar. The results of the PINAX statistical tests are similar to those displayed in Table 1.

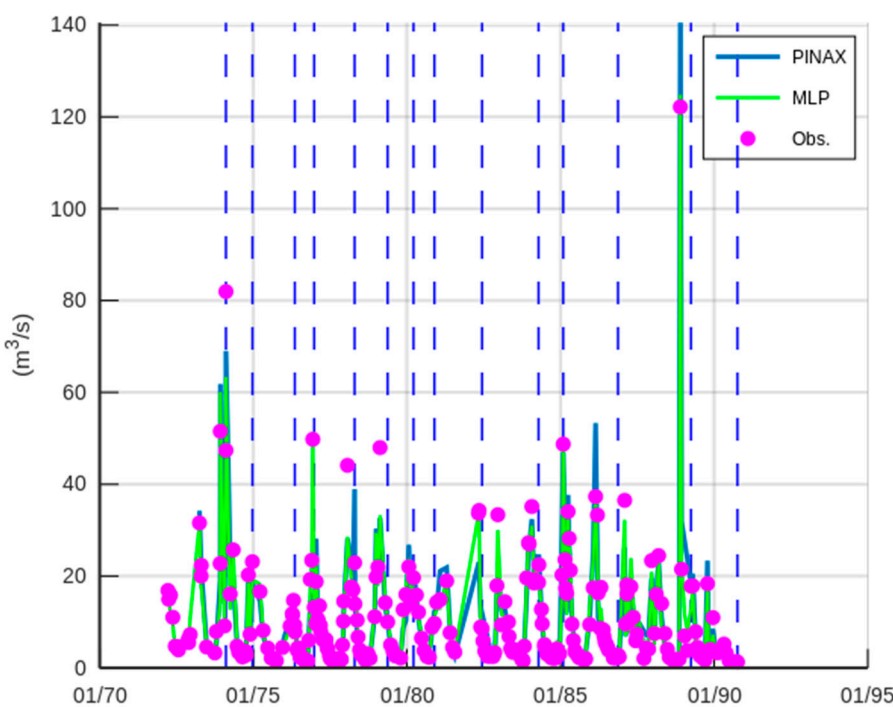

**Figure 9.** Trikeriotis case study—discharge calculated with the MLP and discharge calculated with the 14 PINAX-obtained regressions. The start and end of these 14 periods are marked with the dashed vertical lines.

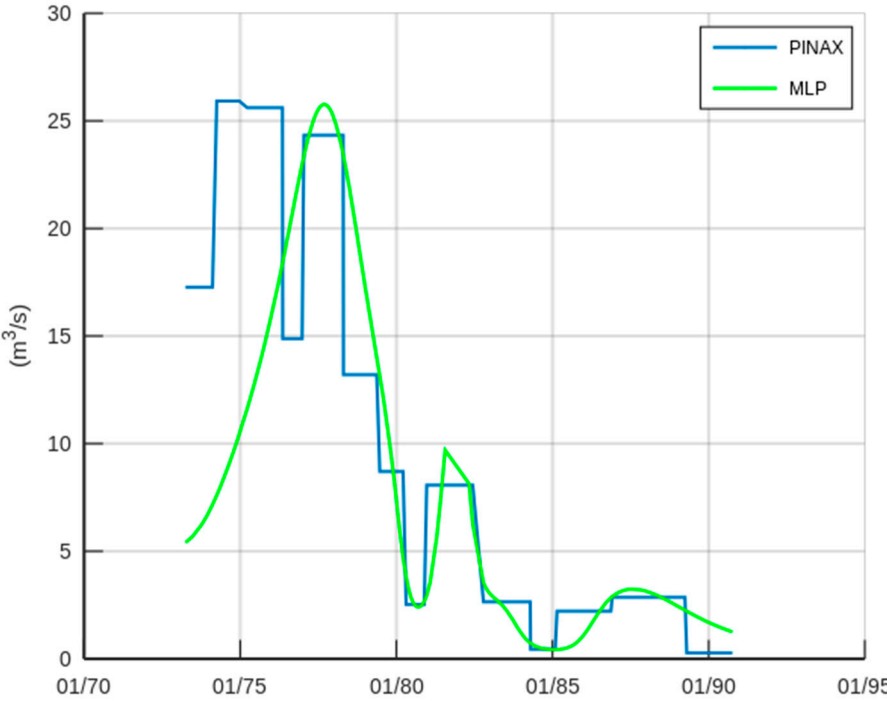

**Figure 10.** Trikeriotis case study—discharge over time corresponding to a constant stage (equal to mean observed stage) as it is calculated by the 14 PINAX-obtained regressions and the MLP.

### 3.3. Case Study—Agrafiotis River

The dataset consists of 243 stage–discharge measurements from Agrafiotis River, Greece. The measurements extend over a twenty-four-year period with the first one performed on 8 August 1966 and the last one on 26 November 1990. This was a

challenging case study since PINAX filtered 86 out of the 243 measurements. PINAX partitioned the available stage–discharge measurements into 16 periods. The DC of the combined corresponding 16 regressions was 0.96.

In this case study, like the previous, an MLP was applied directly on the whole dataset. The topology was identical with that used in the previous case study (Trikeriotis River). The DC of the MLP for the whole dataset (the DC of PINAX is for the dataset without the outliers) was 0.96. The fit of the 16 PINAX-obtained regressions to the filtered dataset and of the MLP to the whole dataset is displayed in Figure 11.

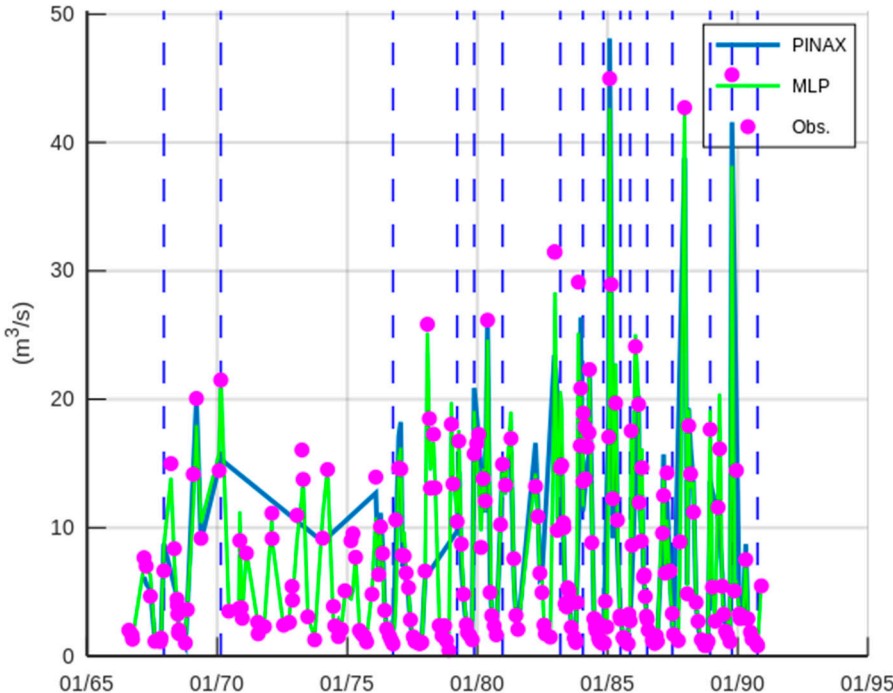

**Figure 11.** Agrafiotis case study—discharge calculated with the MLP and discharge calculated with the 16 PINAX-obtained regressions. The start and end of these 16 periods are marked with the dashed vertical lines.

Figure 12 displays the discharge calculated by the MLP and the PINAX-obtained regressions for a constant stage equal to the mean stage. According to this figure, the cross-section changes estimated by the two approaches are only roughly similar. However, this is justified by the fact that the MLP used the whole dataset, whereas PINAX used a portion (65%) of the dataset. The results of the PINAX statistical tests are similar to those displayed in Table 1.

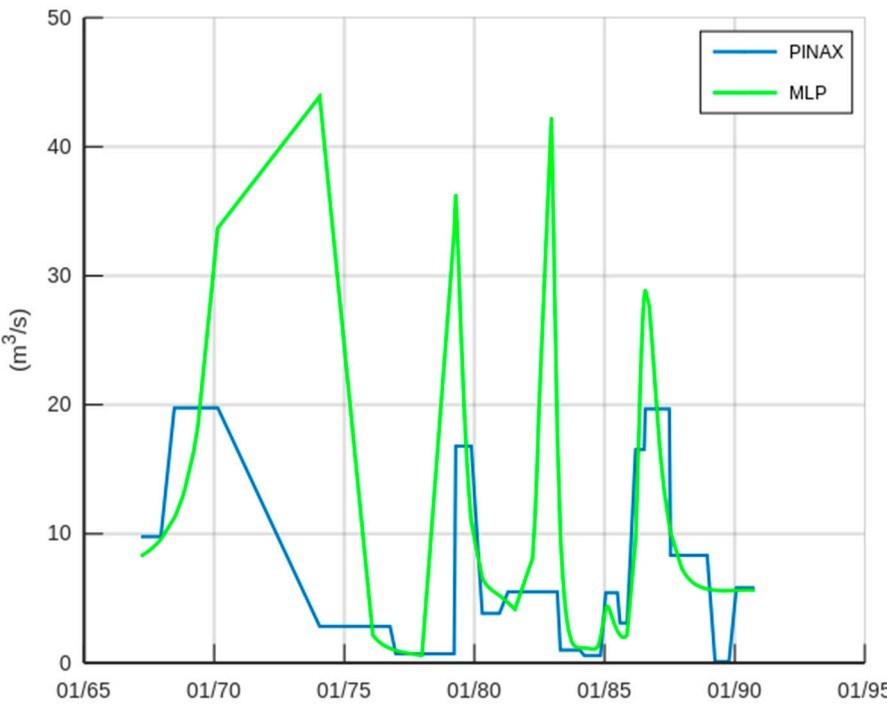

**Figure 12.** Agrafiotis case study—discharge over time corresponding to a constant stage (equal to mean observed stage) as it is calculated by the 16 PINAX-obtained regressions and the MLP.

## 4. Discussion and Conclusions

In this study, we tested the application of machine learning approaches to the development of rating curves. Initially, a clustering method, DBSCAN, was employed to partition the data into groups in order to follow the standard approach (to fit a power–law trend line). However, the performance of this approach was inferior to that of earlier statistical approaches. For this reason, the contribution of DBSCAN only for filtering outliers before the application of an MLP network was evaluated in the first case study. Though DBSCAN filtering in the first case study appeared to be hydraulically consistent, the last two case studies, where MLP was applied to the whole dataset without filtering, exhibited remarkable performance, thus, questioning the necessity of prior filtering. Ideally, the outliers obtained by the prior filtering should be explained on a hydraulic basis (e.g., "This is obviously an outlier because..."). Otherwise, filtering may be omitted.

The lesson learnt from this study is that an MLP network can be applied to approximate the stage–discharge relationship, even without prior filtering of outliers (though an increased number of outliers, i.e., erroneous measurements, could create issues with overfitting, see further down). MLP networks have been previously used by other researchers. The novelty of our study was that we included time as an input along with the stage to reproduce the time-variant stage–discharge relationship (see Equation (2)). The inclusion of time as input allowed the application of MLP with a fictitious constant stage in order to obtain a schematisation of the cross-section change. A fixed cross-section would result in a flat line over time (the same discharge for the same stage). However, if material is deposited, then the discharge, for the same stage, will decrease. Therefore, high values in Figures 8, 10, 12 correspond to periods after degradation and low values to periods after aggradation. The applications to the three case studies indicated that both the machine learning and the statistical approaches presented similar schematisation of the cross-section change. This is an additional verification of the machine learning approach. For example, unrealistically high discharges in Figures 8, 10, and 12 would indicate an overfitting of the MLP network. An overfitting would result in

increased influence of the measurements that are erroneous (in case outliers are not filtered).

The advantage of an approach based on machine learning is its simplicity and applicability. The involved tools and processes are routinely used in the field of modern statistics. Regarding the practitioners, this methodology can be implemented in a spreadsheet with the addition of external packages provided by third party vendors [35,36], which makes it more attractive compared to sophisticated statistical approaches. A disadvantage of the approaches based on machine learning is the time required to train the network. In the three case studies, the training time was half an hour. Another disadvantage is that the suggested methodology requires some expert evaluation to avoid overfitting in cases where a significant portion of the measurements is erroneous. However, this involvement of the user may have the benefit of gaining a better insight into the processes of the studied hydrological system.

**Author Contributions:** Conceptualization, E.R. and D.K.; methodology, E.R.; software, E.R. and D.K.; validation, J.L. and D.K.; formal analysis, E.R.; investigation, E.R.; resources, J.L.; data curation, D.K.; writing—original draft preparation, E.R.; writing—review and editing, J.L. and D.K.; visualization, E.R.; supervision, J.L.; project administration, E.R.; funding acquisition, E.R. All authors have read and agreed to the published version of the manuscript.

**Funding:** This research was funded by the Internal Grant/Award of National Observatory of Athens "Low computational burden flood modelling in small to medium-sized water basins in Greece"— 5080. The APC was funded by the Internal Grant/Award of National Observatory of Athens "Low computational burden flood modelling in small to medium-sized water basins in Greece"—5080.

**Institutional Review Board Statement:** Not applicable.

**Informed Consent Statement:** Not applicable.

**Data Availability Statement:** The datasets analysed in this study can be provided upon request. A tool will be made publicly available upon publication at http://hydronoa.gr (accessed on 1 September 2022).

**Conflicts of Interest:** The authors declare no conflict of interest.

## Appendix A

The following images display an attempt to partition the stage–discharge measurements of Sakoulevas River by employing 10 clustering tools.

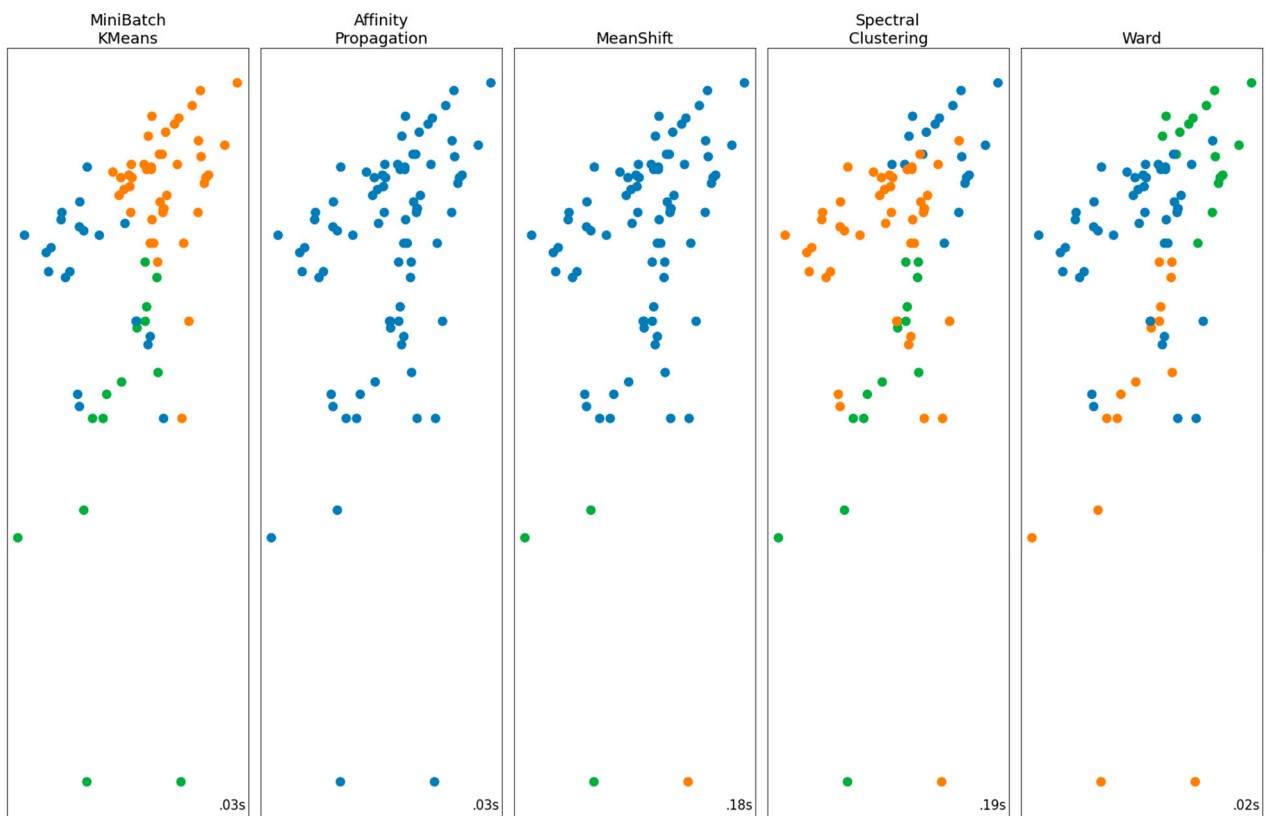

**Figure A1.** Attempt to partition the stage–discharge measurements of Sakoulevas River with the clustering methods: MiniBatch Kmeans, Affinity Propagation, MeanShif, Spectral Clustering, and Ward. Different colours indicate different classes. Black dots correspond to outliers. The *x*-axis is the discharge, the *y*-axis is the stage.

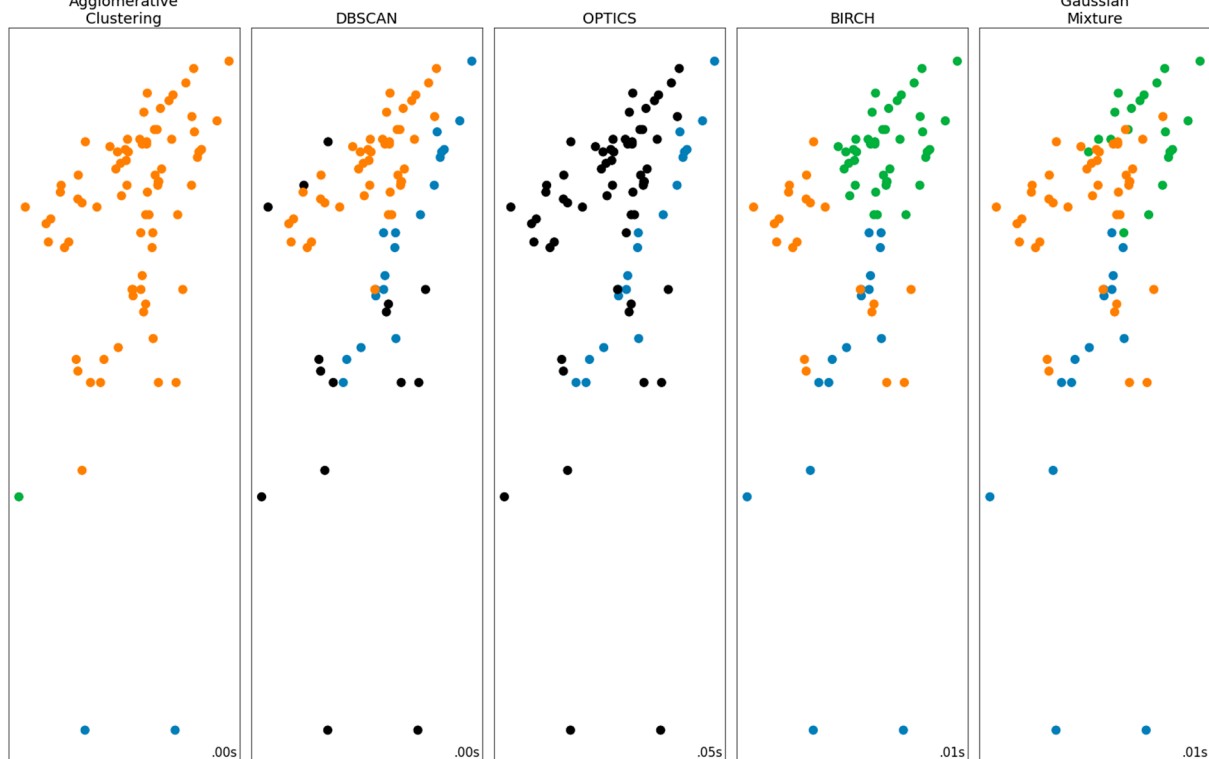

**Figure A2.** Attempt to partition the stage–discharge measurements of Sakoulevas River with the clustering methods: Agglomerative Clustering, DBSCAN, OPTICS, BIRCH, and Gaussian Mixture.

Different colours indicate different classes. Black dots correspond to outliers. The *x*-axis is the discharge, the *y*-axis is the stage.

**Appendix B**

The following figure displays the regression analysis on the two groups of stage–discharge measurements obtained with DBSCAN.

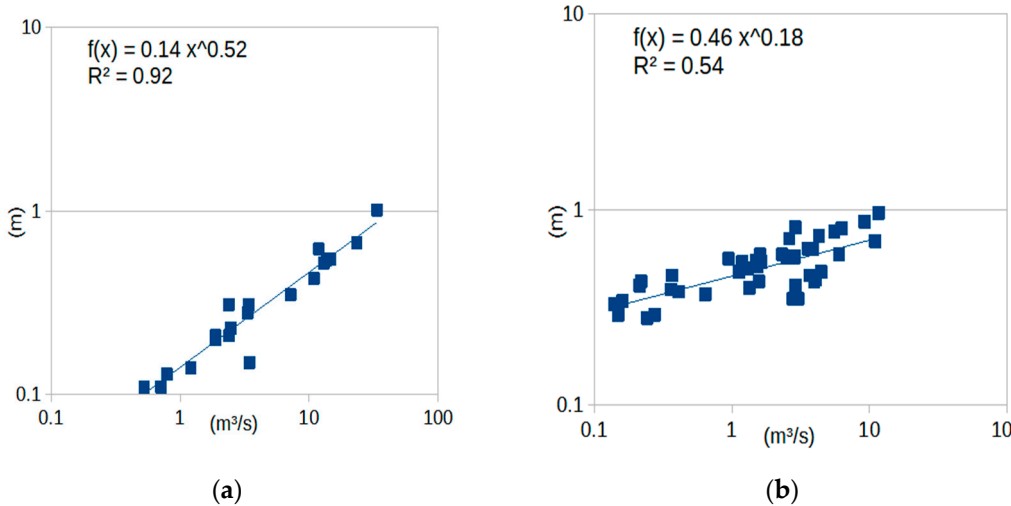

(**a**) (**b**)

**Figure A3.** The fit of a power law trend line to data partitioned with DBSCAN: (**a**) period 1 (02/1964–04/1966); (**b**) period 2 (05/1966–01/1972).

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
