# Peer review of "Development of Rating Curves: Machine Learning vs. Statistical Methods"

_hydrology, doi:10.3390/hydrology9100166_

Round 1
Reviewer 1 Report (Previous Reviewer 1)
The manuscript “Development of rating curves: machine learning vs statistical methods” presents a machine learning method to obtain and use time-varying stage-discharge relationships that capture channel geometry changes. They demonstrate and explain the technique using three case studies and compare their results to the standard PINAX method.
The paper is organized well and communicates both approach and findings clearly. With current machine learning libraries, this approach is accessible to practitioners in the field.
I recommend accepting the manuscript.
I have included a markup of the manuscript with some minor typos and grammar suggestions.
Specific Comments
L77 Need to define SVM

Author Response
Dear Reviewer, thank you for your positive comments. We have addressed your suggestions in the annotated pdf file. For some of these comments, detailed answers have already been given. We did upload them to the editorial system, but apparently they were not delivered to you (the other two reviewers appear also unaware of our replies). Please find them in the attached pdf file.

Reviewer 2 Report (Previous Reviewer 2)
Streamflow measurements provide the most valuable hydrological information, but at the same time, are the most difficult to obtain. For this reason, discharge records of regular intervals are usually obtained indirectly by a stage-discharge rating curve, which establishes a relation between measured water levels to a volumetric rate of flow. This manuscript employed a simple technique to take into account the dependency of the stage-discharge relationship on time: let time along with stage be the machine learning model inputs. The topic is interesting, and the main conclusions are well supported by results. I would recommend a Moderate Revision.
(1) Abstract: More quantitative results should be provided.
(2) Introduction: A better introduction of climate change should be provided. For example, why climate change would change precipitation and streamflow. The following references may help.
Yin J B, Guo S L, Gentine P, Sullivan S C, Gu L, He S, Chen J, Liu P. 2021a. Does the hook structure constrain future flood intensification under anthropogenic climate warming? Water Resour Res, 57: e2020WR028491
Prein A P, Rasmussen R M, Ikeda K, et al. 2017. The future intensification of hourly precipitation extremes. Nat Clim Chang, 7: 48-52
(3) Methods: The main differences between different methods should be further highlighted.
(4) Results: Some deep exploration about the calculation results should be presented.
(5) Discussion: please provide the main limitation and future works, particularly application in data-sparse regions.
(6) The language can be polished.
Author Response
Dear Reviewer, we have added the reference of Yin et al. Regarding the rest of your comments, detailed answers have already been given for these comments. We did upload them to the editorial system, but apparently they were not delivered to you (the other two reviewers appear also unaware of our replies). Please find them in the attached pdf file.

Reviewer 3 Report (Previous Reviewer 3)
Major remarks:
- Could you explain why you selected DBSCAN as a final clustering method? From Figures A1 and A2, it is not obvious to me (some clusterings give desired 3 groups). I propose to add some measures of the quality of clustering (eg. CH index or silhouette coefficient).
- I propose to add (or change one of the methods) HDBSCAN - an improved variant of DBSCAN.
- Why did you use MLP for such data? You have a few observations (especially when you removed outliers). In my opinion, you could check more methods of regression machine learning.
Minor remarks:
- I propose alphabetical order of references.
Author Response
Dear Reviewer, detailed answers have already been given for these comments. We did upload them to the editorial system, but apparently they were not delivered to you (the other two reviewers appear also unaware of our replies). Please find them in the attached pdf file.

Round 2
Reviewer 2 Report (Previous Reviewer 2)
I am happy with the revision. It can be accepted.
Reviewer 3 Report (Previous Reviewer 3)
The authors made an effort to correct and improve their paper. They responded to all raised problems and I appreciate this.
This manuscript is a resubmission of an earlier submission. The following is a list of the peer review reports and author responses from that submission.
Round 1
Reviewer 1 Report
The manuscript “Development of rating curves: machine learning vs statistical methods” presents an interesting method to develop time-dependent rating curves. As channel geometries change because of floods, deposition, or other processes, the rating curve can change. It is not always clear when the channel has changed significantly enough to require a new rating curve. This approach uses only available data.
The approach is interesting, but I recommend some additional work and analysis before publication.
1) Demonstrate on more than 1 data set. It is important to apply new methods to more then one data set as the data set may be unique. It would be important to see how the method performs on other data. It would be particularly good if a data set were used where a known change in channel geometry occurred to see if the method identified it.
2) A better description of the method – there are several issues where the manuscript is contradictory
3) Better communication of the groups. Different colors are useful, but other methods might be beneficial – especially for time-series groups (which are the most important) and better presentation of resulting rating curves.
4) The time varying rating curve (Figure 8) shows “continuous”, sometimes very large changes. For example in PINAX period 1. I would expect most large changes in channel geometry to occur abruptly. Some things, like sedimentation, could cause graduate changes over time. I think this might be related to the way the time data were normalized. See below
Time data. You used z-score normalization, but time data are not normally distributed. In fact they are “pure” trend. If you subtract the trend, you would get zero. I suggest you try normalizing the time data with a min-max normalizer (e.g., go from 0 to 1 over the period of study).
Outliers
in lines 183 – 185 you state the outliers were not removed; in the remainder of the paper, you state they were removed. This needs to be clarified.
In Figures 3 and 4, the end of the time series is marked as outliers. Just looking at the plots, these seem to be a new group, rather than outliers. Was there an event about 1971 that changed the channel geometry? You attempt to provide reasons that these data are outliers, but if this is a new geometry, these arguments do not hold. The fact that they are continuous in time, suggests that they are not outliers.
Since there is only one data set, you should have more information on channel geometry. Use satellite images, notes, or other to see if you can identify any changes that match the data and discuss.
Rating Curve
I would expect than many hydrologists would be skeptical of a rating curve that changed as constantly as that shown in Figure 8. I would expect the curve to have “regions” where it is relatively constant, such as that associated with PINAX Period 2. The large changes associated with PINMAX Period 1 do not seem realistic. I would not expect geometry changes that follow that curve. This needs to be discussed in detail and reasons given to accept these large continoius changes, rather than the abrupt changes we would expect. Having more data sets would better support these claims.
Parameters
It would be interesting to see the importance of the time parameter and how it changed for different time periods in the data.
I recommend that the author address the comments and add two or more demonstration/validation data sets before publication.
I am recommending rejection with a strong encouragement to resubmit.
Specific Comments
L40 Delete the parenthetical phrase “tipping buckets”
L51-62. These discussions are changes in precipitation or flow regime, not geometry changes that would result in changes to the rating curve. If the rating curve is accurate and geometry does not change, then these process changes should not affect the rating curve.
L64-84 Again, the discussion does not clearly separate methods related to a rating curve (correlation between stage and flow) and regimes of flow and stage. This is a good presentation, but need to be better communicated.
L140 Need more than one data set. Should compare your rating curve to the accepted curve for this station.
L149 – 152 Should use min-max scaling for time, not z-score. Data are not normally distributed.
L184 States that MLP model included outliers – so why use DBSCAN if you are not excluding outliers.
L184-186 I think NOT excluding outliers is good. Looking at the plots, the “outliers” at the end of the time period, do not seem to be outliers, but a new channel geometry.
Figuires 3, 4, 5 That all the points, other than 4 and 14, present as a different group, rather than outliers in Figure 5. Pints 1 and 2, and 3 and 4 are at different points in the time line, so exclusion might make senses. However, in Figure 3 and they look reasonable. I don’t see point 3 in Figure 4.
Figure 5. A linear regression or polynomial regression fit to the three groups would help better visualize the data. For this you might want to exclude points prior to point 6, as they are from a different part of the time line.
Figure 6 It would be helpful to have vertical lines or colors showing the limits of the various groups.
Figure 6 Both MIP and PINAX estimate significantly higher flows than the existing rating curve for the flood in 1965. IT would be good to present the existing rating curve and your rating surve to see the difference. You might have to present your curve as a set of curve in time as it continually changes.
Figure 7 Linear or polynomial fit lines for the different groups would be helpful.
L273 Were outliers excluded or not?
L277 – 283, this is an interesting discussion
L297 – 304 It isn’t clear why this was included and it is difficult to follow. It seems to suggest that you got luckly with your 1st MLP design, that if you had chosen another one, it wouldn’t have worked. This is why it is important to demonstrate/validate on more than one data set. Would another data set work better with a 1-20-1 design?
L310-311 Were outliers filtered or not?
L310-311 I think it would be useful to use DBSCAN or other to create “time periods” and then fit different MLP models to the period.
L320-328 I would like more implementation details – maybe in the appendix.
Reviewer 2 Report
Rating curves are hard to develop because they require simultaneous measurements of discharge and stage over a wide range of stages. Furthermore, the shear forces, especially during floods, change the streambed shape and roughness. As a result, over long periods, the stage-discharge measurements tend to form clusters, to which different rating curves apply. For the identification of these clusters, various robust statistical approaches have been suggested by researchers, which, however, have not become popular among practitioners because of their complexity. This manuscript examined the advantages of a very simple technique: use time as one of the machine learning model inputs, and use unsupervised learning for filtering outliers. The topic is interesting and within the scope of Hydrology. I would recommend a Major Revision.
(1) Abstract: Please provide some quantitative results.
(2) Introduction: Authors should highlight the main differences of previous studies, i.e., machine learning and statistical methods. It is also important to highlight the main potential novelty.
(3) Materials and Methods: There are so many statistical methods. Why authors only use the PINAX?
(4) Section 2.2: There are numerous machine learning methods. It is important to clarify that why you use the selected one.
(5) I would suggest omitting those sentences about Excel, Matlab…
(6) Results: This section is not clearly clarified. More analysis should be provided.
(7) The English should be polished.
(8) Conclusions: I would suggest discuss potential application in data-sparse area by using remote sensing data. The significance of this study in a warming climate is also important. The following reference may be helpful.
Sunilkumar, K., Narayana Rao, T., Satheeshkumar, S., 2016. Assessment of small-scale variability of rainfall and multi-satellite precipitation estimates using measurements from a dense rain gauge network in Southeast India. Hydrol. Earth Syst. Sci. 20, 1719–1735.
Yin et al. 2021. Does the hook structure constrain future flood intensification under anthropogenic climate warming?. Water Resources Research 57: e2020WR028491.
Berghuijs, W. R., Harrigan, S., Molnar, P., Slater, L. J., & Kirchner, J. W. (2019). The relative importance of different flood-generating mechanisms across Europe. Water Resources Research, 55(6), 4582–4593.
Nie, J., Ruetenik, G., Gallagher, K., Hoke, G., Garzione, C.N., Wang, W., Stockli, D., Hu, X., Wang, Z., Wang, Y., Stevens, T., Daniˇsík, M., Liu, S., 2018. Rapid incision of the Mekong River in the middle Miocene linked to monsoonal precipitation. Nat. Geosci. 11, 944–948
Yin JB, Guo SL, Gu L, et al. 2021. Blending multi-satellite, atmospheric reanalysis and gauge precipitation products to facilitate hydrological modelling. Journal of Hydrology593, 125878.
Reviewer 3 Report
Major remarks:
- Could you explain why you selected DBSCAN as a final clustering method? From Figures A2 and A3, it is not obvious to me (some clusterings give desired 3 groups). I propose to add some measures of the quality of clustering (eg. CH index or silhouette coefficient).
- Could you write more about calibrating parameter ε in the DBSCAN method?
- I propose to add (or change one of the methods) HDBSCAN - an improved variant of DBSCAN.
- Why did you use so old data set? In my opinion, this paper could benefit a lot from the better (new) data set.
- Manual looking for outliers is interesting. However, you employed DBSCAN to select outliers and you didn't use clustering results in MLP. It is OK, but I recommend using methods dedicated to looking for outliers like Isolation Forests or Local Outlier Factor.
- Why did you use MLP for such data? You have a few observations (especially when you removed outliers). In my opinion, you could check more methods of regression machine learning.
- The general mathematic framework for R-squared (coefficient of determination) doesn't work out correctly if the regression model is not linear. I propose using other quality measures for regression: RMSE, MSE, MAPE, or MAE. When we compare models with more than one regressor we should use the corrected coefficient of determination.
Minor remarks:
- I think that Figure A1 requires some additional explanation.
- I propose alphabetical order of references.